# Validity of Domain-Specific Sedentary Time Using Accelerometer and Questionnaire with activPAL Criterion

**DOI:** 10.3390/ijerph182312774

**Published:** 2021-12-03

**Authors:** Rina So, Tomoaki Matsuo

**Affiliations:** 1Ergonomics Research Group, National Institute of Occupational Safety and Health, Kawasaki 214-8585, Japan; matsuo@h.jniosh.johas.go.jp; 2Research Center for Overwork-Related Disorders, National Institute of Occupational Safety and Health, Kawasaki 214-8585, Japan

**Keywords:** sedentary behavior, sitting, validation study, worker, sedentary behavior, physical activity

## Abstract

Accelerometers based on the cut-point method are generally the most used in sedentary time (ST) research. However, mixed cut-points are an issue, so an accelerometer based on metabolic equivalents (METs) could be used as an alternative. This study aimed to validate a METs-based accelerometer (HJA-750C, OMRON) and a questionnaire that estimates domain-specific sedentary time measures using activPAL as a criterion value. We also examined whether measurement validity differed according to gender and occupation. We used data from 242 workers in the validation study. Participants wore activPAL on the thigh and OMRON on the waist for seven consecutive days with daily recording logs. The Workers Living Activity-time Questionnaire (WLAQ) was administered once. The domain-specific ST assessed quantities of ST during commuting, working time, non-working time on a workday, and non-workday. Intraclass correlation coefficients (ICC) and Spearman’s rho coefficients were then used to conduct analyses. The OMRON accelerometer showed acceptable values (*r* = 0.67–0.86 and ICC of 0.63–0.87) in the overall domain-specific ST. Additionally, each measurement result suggested that working time is the most accurate domain to measure ST (ICC of 0.87 for OMRON and 0.68 for WLAQ). Moreover, there were no differences in the overall validity of the results according to gender and occupation. The METs-based accelerometer has acceptable validity for ST measurements to be used among workers. Additionally, working time may be the preferred domain for the accurate assessment of ST in both objective and subjective measurements. These results can advance the quality of the sedentary research field.

## 1. Introduction

Due to advancements in automation and technology, sedentary behaviors can increasingly manifest habitually [1], especially in the working generation [2]. Several studies [3,4] have consistently shown that sedentary time (ST), independent of physical activity (PA), is associated with an increased risk of cardiovascular mortality and metabolic disorders. These observations point to the importance of accurately measuring sedentary behaviors to understand the role of ST in health outcomes.

To date, a thigh-worn activPAL (PAL Technologies Ltd., Glasgow, UK) with an inclinometer sensor is the gold standard for its accuracy in assessing time spent sitting or lying down [5]. For other objective measurements, various accelerometers that estimate ST are frequently used in epidemiological studies [6]. For instance, ActiGraph (ActiGraph, Pensacola, FL, USA) is the most used triaxial accelerometer in the ST research field [7]. It classifies activity using specified count thresholds, which is known as the cut-point method, and maintains a high degree of freedom for researcher processing (i.e., modification of cut-point, equations, and analysis). However, though ActiGraph has shown various cut-off points, previous studies have shown differing results; hence, it is unclear which cut-off point is correct [8,9]. Moreover, studies based on activPAL have shown variability in the validity observed between studies [9,10,11].

In contrast, the OMRON triaxial accelerometer (HJA-750C, OMRON Healthcare, Kyoto, Japan) is widely used in the study of PA [12,13], and the validity of its metabolic equivalents (METs) estimation has been confirmed using the Douglas bag method [14]. OMRON has a proprietary algorithm for estimating METs every 60 s using a built-in triaxial accelerometer directly as raw data. It then calculates METs without the need for a researcher to decide the cut-off point. However, each accelerometer uses a different algorithm; hence, time estimation may differ between accelerometers even in the same sitting. Furthermore, the validity of OMRON in distinguishing ST based on activPAL remains unknown.

While the majority of research assessing ST uses objective technology with lower measurement error [15,16], most validity studies focus on specific or total ST [5,8,17]. Generally, ST occurs across numerous domains, and each period of domain-specific ST may affect health risks differently [18,19]. Therefore, the questionnaire is an effective tool for assessing ST within various domains, especially in large-scale epidemiological research with a lower participant burden. Several questionnaires have been developed to measure ST, and their validity was reviewed [20]. However, given that it is not always possible to assess domain-specific ST [21,22], the validity of the questionnaires in assessing domain-specific ST with objective measures remains to be examined.

Thus, the Worker’s Living Activity-time Questionnaire (WLAQ) was developed primarily to measure ST within four specific domains: commuting time, working time, non-working time on a workday, and non-workday [23]. Matsuo et al. evaluated WLAQ and demonstrated good validity using activPAL as a criterion [23], and a recent meta-analysis reported the high quality of this study [20].

Knowledge about the difference between measurement tools and the validity of domain-specific ST is helpful in correctly interpreting results in this field of research. Therefore, we aimed to evaluate the validity of a METs-based accelerometer (OMRON, HJA-750C) using activPAL as a criterion value. We then compared the validity of domain-specific ST, measured by OMRON and WLAQ with activPAL, and examined whether measurement validity differs according to gender and occupation.

## 2. Methods

### 2.1. Participants

We recruited a total of 332 participants through a website advertisement. We included them based on the following criteria: (1) age (30–60 years), (2) whether they lived in the Tokyo area, and (3) whether they were full-time workers for at least four days a week. We explained the details of the study (i.e., aims and design) to each participant before obtaining written informed consent. This study was conducted following the guidelines of the Declaration of Helsinki, and its protocol was reviewed and approved by the Ethical Committee of the National Institute of Occupational Safety and Health, Japan (ID H2744).

### 2.2. Data Collection

The participants visited our laboratory on two occasions, one week apart. On the first visit, they underwent anthropometric measurements and completed sociodemographic and WLAQ questionnaires. Afterward, the participants received instructions on how to wear the activPAL and OMRON devices. They were then directed to record daily log-times during the day (i.e., workday or non-workday, bedtime and wake-up time, time of start and end of work, and typical or unusual workdays), and any periods they may not have worn the devices.

We requested them to wear the activPAL for 24 h and OMRON for all waking hours, except when sleeping and during water-based activities (e.g., bathing, showering, and swimming). Further, we requested that they maintain their usual activities during a measurement period of seven consecutive days. Each participant agreed to a second visit, on the eighth day of monitoring, to return the devices, confirm compliance, and share any wearing issues.

### 2.3. Measures

#### 2.3.1. Anthropometric and Sociodemographic Measures

We measured height (to the nearest 0.1 cm) and body weight (to the nearest 0.1 kg) during the first visit. We calculated body mass index (BMI) as the weight in kilograms divided by the square of their height in meters. Additionally, occupational information such as employment type (whether they are regular staff, temporary workers, contract employees, and entrusted employees), industry type, and presence or absence of shift work were collected using a questionnaire.

#### 2.3.2. Assessment of Sitting Time by Devices

The activPAL3^TM^ is a small, light inclinometer that continuously records time spent lying, sitting, standing, stepping, and non-wear time in 15-s intervals. The device was waterproofed using a nitrile sleeve and fitted directly to the mid-thigh using 3 M Tegaderm^TM^ tape. We provided an instruction leaflet and extra tape to adjust and reattach the device if it was uncomfortable or irritating. The detailed time data (from 0:00 to 24:00 h) on each measurement day were exported into a CSV file using activPAL software (version 8.11.2). Non-wear time was based on information from a daily log and visual inspection of the CSV data.

OMRON accelerometer (HJA-750C) is one of Japan’s most popular validated triaxial accelerometer activity monitoring devices [12,13] that accurately measures the frequency, intensity, and duration of PA [14]. This device was worn on the waist, and METs data were collected in 60-sec intervals. Sedentary behavior was defined with an estimated accelerometer intensity of ≤1.5 METs [24] by default. The CSV files from the device were downloaded by OMRON health management software BI-LINK for PA professional edition version 2.0. Then, the files were processed using a custom-written Excel macro program for compiling data. The macro program automatically calculated each participant’s domain-specific average time (min) spent doing sedentary behavior of ≤1.5 METs, light-intensity PA of 1.5–2.9 METs, moderate-intensity PA of 3.0–5.9 METs, and vigorous-intensity PA of ≥6.0 METs using both the OMRON data and daily log information. Non-wear time was defined as at least 60 consecutive minutes of 0 counts per minute.

#### 2.3.3. Assessment of Sitting Time by the Worker’s Living Activity-Time Questionnaire

The WLAQ [23] with acceptable validity includes questions on when individuals perform certain activities (e.g., going to bed, waking, leaving the house, and arriving at and leaving the workplace) based on the monthly recall period. These questions make it possible to calculate working hours, leisure time, and sleep time. In fact, once we learn the proportional time the participants spend sitting, we can calculate the actual number of minutes per day they spend sitting, walking, or standing during typical periods in a worker’s life: (a) commuting time, (b) working time, (c) leisure time on a workday, and (d) leisure time on a non-workday.

### 2.4. Data Procedure and Statistical Analyses

Based on daily log information, if we found an unusual workday recorded, such as a business trip or a half-day off, or if the participants failed to record the required time, we removed the data from the average calculation. To be eligible, participants needed to wear the accelerometer for at least four workdays and one non-workday, with the average wearing time on workdays being ten hours or more [25]. Total wearing time per day was calculated using OMRON after removing sleep and non-wear time at one-minute intervals with zero consecutive counts for at least 60 min [26]. The individual’s average ST during the non-working time on a workday was calculated from ST in the morning (from waking to leaving the house) and evening (from arriving home to going to bed).

We excluded 23 participants to equalize living activity time for statistical purposes and 67 participants due to a lack of data collection with the activPAL or OMRON. In addition, 31 participants who did not meet the average wearing time on workdays (of ten hours or more) were excluded. Consequently, we used data from the remaining 211 participants in our final analyses. To evaluate the sub-analysis, we classified occupations as either active or sedentary work. Active work included occupations in sales (*n* = 19), manufacturing (*n* = 6), cleaning (*n* = 10), healthcare (*n* = 1), teaching (*n* = 1), and occupations in the plastic sector (*n* = 5); meanwhile, sedentary work included occupations in the clerical (*n* = 98) field. We excluded occupations that included sales planning (*n* = 41), technical support (*n* = 28), and transportation (*n* = 2), as we considered them too ambiguous to have high or low occupational PA to interpret the results.

We calculated continuous data as mean ± standard deviation (SD) and used the Student’s unpaired *t*-tests to compare differences between genders and occupations. Next, we calculated differences between the two devices (activPAL and OMRON) and WLAQ using analysis of variance (ANOVA). We then calculated intraclass correlation coefficients (ICCs) to compare the validity of each domain-specific ST measured by activPAL, OMRON, and WLAQ.

ICC were interpreted as poor (<0.40), fair to good (0.40–0.75), or excellent (>0.75) [27]. We performed Spearman’s rho (ρ) coefficients to determine the association and bivariate correlation coefficients between each method for ST. We then interpreted the ρ values to indicate the strength of the correlations as follows: <0.30 as weak, 0.30–0.49 as low, 0.50–0.69 as moderate, 0.70–0.89 as strong, and ≥0.90 as very strong [28]. Finally, we used Bland–Altman plots to assess bias visually [29]. All statistical analyses were performed using SAS version 9.4 (SAS Institute Japan, Tokyo, Japan). Results were considered significant at *p* < 0.05.

## 3. Results

The 211 participants included 112 men (age: 47.1 ± 7.3 years, BMI: 24.0 ± 3.0 kg/m^2^) and 99 women (age: 46.5 ± 7.4 years, BMI: 21.6 ± 2.8 kg/m^2^). The average daily OMRON wearing time of each domain-specific period was as follows: 56.7 ± 31.2 min commuting, 530.5 ± 91.7 min working, and 308.1 ± 105.5 min non-working time during a workday, and 669.4 ± 183.9 min during a non-workday. In addition, the average daily activPAL wearing time of each domain-specific period was as follows: 69.2 ± 36.8 min commuting, 608.5 ± 94.5 min working, and 414.5 ± 98.8 min non-working time during a workday, and 990.8 ± 147.2 min during a non-workday. The gender differences of domain-specific ST are also shown in Table 1. The ST during working time and non-workday was markedly shorter in women than in men (*p* < 0.01), and there were partial differences in commuting and non-working time on workday between the genders.

ST in each domain-specific period measured by the three instruments (i.e., activPAL, OMRON, and WLAQ) is presented in Table 1. There were significant differences between the averages measured by each instrument in all domain-specific ST (*p* < 0.01), except in ST during working time in men (*p* = 0.17). Further, all instruments showed that women had shorter ST than men during working time (*p* < 0.01) and non-workday (*p* < 0.01), but significantly longer ST during non-working time on workdays (*p* < 0.01).

Table 2 shows the validity of OMRON and WLAQ compared to the activPAL in each domain-specific ST. ICC values ranged from 0.61 to 0.87 for OMRON, indicating good to excellent validity for ST in each domain-specific period and especially excellent validity for working time ST. ICC values of WLAQ ranged from 0.35 to 0.58, which indicated good validity for most domain-specific ST (e.g., during the commuting, working, and non-working time on a workday). In contrast, the ICC for the non-workday (0.12) exhibited a poor correlation. Spearman’s ρ values ranged from 0.68 to 0.87 for OMRON and from 0.28 to 0.68 for the WLAQ. Moreover, OMRON showed a strong correlation across all domain-specific STs, except for a moderate validity on non-workdays; Spearman’s ρ values for WLAQ were lower than those of OMRON. In addition, validity on non-workdays with both OMRON and WLAQ was lower than in other domain-specific periods. Last, we found a similar trend of validity for OMRON and WLAQ in men and women, that is, no gender differences.

Figure 1 shows the Bland–Altman plots comparing each domain-specific ST measured by OMRON and WLAQ, with activPAL as a criterion. The overall plot suggests a reasonable agreement with activPAL, but the limits of agreement of WLAQ were broader than those of OMRON. When comparing activPAL with OMRON, we found no significant mean differences in any domain-specific ST, except during the commuting ST (−6.03 min/day, *p* < 0.01) and non-workday (−29.4 min/day, *p* < 0.01). We also found no significant or proportional biases in all domain-specific ST.

On the other hand, the mean differences between activPAL and WLAQ were 5.97 min/day (*p* < 0.01) in commuting ST (Figure 1E), 21.6 min/day (*p* < 0.01) in leisure-time ST (Figure 1G), and 55.4 min/day (*p* < 0.01) in non-workday ST (Figure 1H). The Bland–Altman plots for activPAL and WLAQ displayed negative correlations during working time ST (r = −0.17, *p* < 0.01), leisure-time ST (r = −0.20, *p* < 0.01), and non-workday ST (r = −0.33, *p* < 0.01). This suggests a significant proportional bias; that is, WLAQ underestimates the high levels of actual ST when it is used for assessment.

The secondary analysis compared each domain-specific ST period according to occupation (Table 3). The results showed that the active work group had significantly less wearing time during the commuting period and less ST during working time than the sedentary group. There were also significant differences between the instruments (i.e., activPAL, OMRON, and WLAQ) in all domain-specific STs (*p* < 0.01), except for ST during the active workers’ working time (*p* = 0.75). Further, there were apparent occupational differences during work only in ST. Table 4 shows the validity comparison between sedentary and active jobs. ICC values for OMRON ranged from 0.49 to 0.77 in sedentary work and 0.70 to 0.87 in active work; both ICC values were good-to-excellent for each domain-specific ST.

## 4. Discussion

This study examined the validity of accelerometer-based and questionnaire instruments for several domain-specific STs using activPAL as a criterion under free-living conditions. To summarize the results, the OMRON accelerometer showed good validity (ρ = 0.67–0.86 and ICC = 0.63–0.87) in overall domain-specific ST. Additionally, the results of all objective instruments (e.g., activPAL and OMRON) and subjective instruments (e.g., WLAQ) examined in this study suggest that working time is the most accurate domain to measure ST. Moreover, no differences were dependent on gender and occupation in the validity results.

Numerous validation studies have shown that activPAL is an accurate instrument that is regarded as the gold standard for measuring time spent doing sedentary activities [5,30]. Nevertheless, an increasing number of studies [6,9] have been using accelerometers for their cost-effectiveness and high degree of freedom in data processing activities. ActiGraph, in particular, has been widely used to assess ST; however, its accuracy has been reported to be low when it measures light intensity PA, including sedentary behavior [9,31], and various cut-off points are also a limitation when considering the best approach to be used [5,10,32].

There were relatively small biases when comparing activPAL with OMRON in our study. It is noteworthy that we found acceptable validity in all domain-specific ST periods, except during the commuting ST (−5.63 min/day, *p* < 0.01). Although the difference between ActiGraph and OMRON (HJA-750C) has not been directly examined, disagreement in estimated values by types of accelerometers may be caused by different algorithms, settings used for estimation behavior, and researcher operability [33]. For example, ActiGraph uses counts converted from their acceleration as raw data, and no equations have been designed to estimate ST. On the other hand, as OMRON estimates METs directly as raw data, analysis based on the definition (estimated accelerometer intensity of ≤1.5 METs) can be performed, which provides more accurate evidence without any cut-point confusion. In addition, a previous study [34] comparing ‘ActiGraphy with an older generation of OMRON (Active Style Pro) showed that OMRON provided more details about different predicted METs in both sedentary and household activity than ‘ActiGraphy. Given these, different accelerometers may give varied results, and a METs-based accelerometer can help enhance the accuracy of ST research. Thus, it is essential to select an appropriate accelerometer for research purposes.

Another notable result is the high validity displayed during working time compared to other domains in OMRON, WLAQ, and activPAL. Recent studies have found that many workers spend considerable amount of ST in their workplace. For example, in a study on large office-based workers, more than 60% of daily ST was accrued during work [35], indicating that the workplace is a significant setting for prolonged bouts of ST [36]. For workers, this behavior at work may be routine, and our results, which showed the best measurement validity during working time, support the previous study. As sitting at work is habitual and restricted to specific periods during the day, a study based on the activity leads to low measurement bias [37]. Moreover, given that unequal domain-specific ST may represent differing associations with health outcomes [18,19], further research on ST during working hours is needed.

Differences between validity by gender and occupation showed a tendency similar to the main results. In the gender comparison, men spent more ST overall than women. A systematic review suggests that various sedentary activities differ by sex [38], and women may lead less structured lives due to the higher prevalence of household and childcare responsibilities upon them. On the other hand, in this study, we tried to validate the ST evaluation of workers based on occupations. As a result, we detected a significant difference in ST during working time by occupational classification.

Additionally, the excellent validity of ST during working time was not affected by occupation. However, due to the variety within some occupational categories, ST and PA levels may be quite heterogeneous within an occupation [39]. Thus, future epidemiological research in this field is needed to understand how variations in sedentary behavior and PA interact to influence health.

This study had several strengths and limitations. Its strengths include a large sample size under free-living conditions using a thigh-worn activPAL inclinometer as the criterion device. Moreover, we simultaneously examined the validity of two different approaches based on high compliance and fully detected daily logs and provided convergent results for domain-specific ST. On the other hand, the accelerometer cannot detect postures; it only estimates the time spent in different PA intensities [40]. Thus, the limitations of each method may degrade the quality and accuracy of the measures. Lastly, the generalizability of our findings is limited to workers, and should be interpreted with care for other populations, including older adults.

## 5. Conclusions

The results of this study suggest that a METs-based accelerometer has acceptable validity for ST measurements in workers. In particular, working time may be the preferred domain for the most accurate assessment of ST. Thus, our findings can be interpreted to help improve the accuracy of future ST research fields and can be beneficial to public and occupational health researchers.

## Figures and Tables

**Figure 1 ijerph-18-12774-f001:**
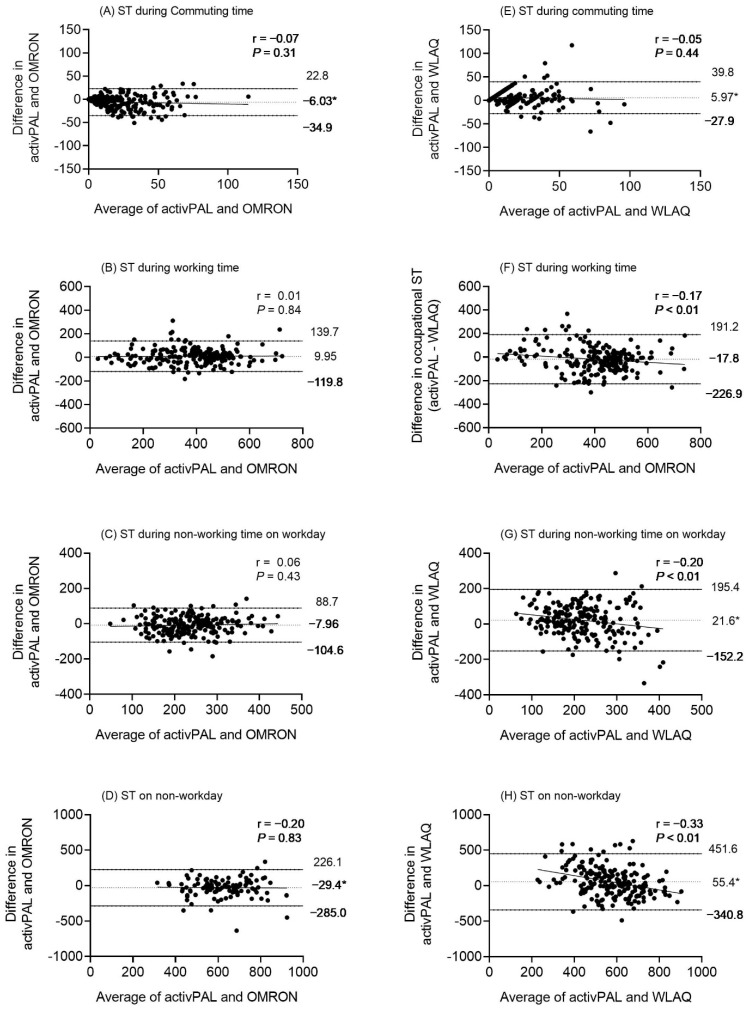
Bland–Altman plot comparing OMRON (left side) and WLAQ (right side) with the activPAL criterion. (**A**,**E**) ST at commuting, (**B**,**F**) ST at working time, (**C**,**G**) ST at non-working time on workdays, and (**D**,**H**) non-workdays. The mean difference is shown as a dashed line, and the 95% limits of agreement appear as solid lines. The correlation coefficients between X and Y are shown. * Significant mean difference (*p* < 0.05).

**Table 1 ijerph-18-12774-t001:** Characteristics of participants and comparison with domain-specific sitting time measured by three measurement methods between the genders.

	Total	Men	Women
	*n* = 211	*n* = 112	*n* = 99
ST during commuting time			
activPAL, min	20.2 ± 20.1	22.1 ± 20.3	18.0 ± 19.5
OMRON, min	26.2 ± 21.0	29.8 ± 21.1	22.1 ± 20.2 *
WLAQ, min	14.2 ± 20.8	15.6 ± 21.2	12.6 ± 20.4
*p* for difference	<0.01	<0.01	<0.01
ST during working time			
activPAL, min	391.2 ± 135.8	443.4 ± 120.9	332.3 ± 127.9 *
OMRON, min	381.3 ± 134.9	431.9 ± 125.8	324.1 ± 121.7 *
WLAQ, min	409.1 ± 152.3	449.0 ± 145.9	364.0 ± 147.4 *
*p* for difference	<0.01	0.17	<0.01
ST during non-working time on workday			
activPAL, min	229.2 ± 75.2	214.4 ± 73.7	246.1 ± 73.6 *
OMRON, min	237.2 ± 72.6	225.6 ± 74.6	250.3 ± 68.3 *
WLAQ, min	207.6 ± 90.4	184.1 ± 84.2	234.2 ± 90.3 *
*p* for difference	<0.01	<0.01	<0.01
ST on non-workday			
activPAL, min	597.4 ± 136.5	612.2 ± 142.8	580.8 ± 127.8 *
OMRON, min	609.6 ± 151.2	630.0 ± 142.1	586.8 ± 158.5 *
WLAQ, min	544.1 ± 188.0	570.8 ± 192.5	514.0 ± 178.9 *
*p* for difference	<0.01	<0.01	<0.01

Values are presented as mean ± standard deviation. Abbreviation: ST, sitting time; WLAQ, The worker’s Living Actiity-time Questionnaire. * Significant differences between sex by Student’s unpaired *t*-tests (*p* < 0.05).

**Table 2 ijerph-18-12774-t002:** Criterion validity of values measured by OMRON and WLAQ compared with activPAL by gender.

	Total (*n* = 211)		Men (*n* = 112)		Women (*n* = 99)	
	ρ	ICC	ρ	ICC	ρ	ICC
Commuting time						
OMRON	0.72	0.70	0.64	0.62	0.83	0.74
WLAQ	0.64	0.60	0.62	0.63	0.65	0.55
Working time						
OMRON	0.87	0.87	0.82	0.83	0.86	0.85
WLAQ	0.68	0.71	0.69	0.66	0.66	0.68
Non-working time on workday						
OMRON	0.73	0.75	0.72	0.71	0.70	0.75
WLAQ	0.44	0.41	0.40	0.35	0.38	0.41
Non-workday						
OMRON	0.68	0.63	0.68	0.61	0.68	0.57
WLAQ	0.28	0.21	0.32	0.26	0.19	0.12

Spearman’s ρ correlations = ρ; Intraclss correlations = ICC. Abbreviation: WLAQ, The worker’s Living Activity-time Questionnaire.

**Table 3 ijerph-18-12774-t003:** Characteristics of participants and comparison with domain-specific sitting time measured by three measurement methods between the occupations.

	Sedentary Work	Activity Work
	*n* = 98	*n* = 42
Age, year	47.3 ± 6.7	48.0 ± 6.8
BMI	22.4 ± 3.1	22.2 ± 2.7
Wearing time, min		
Workday		
During commuting time, min	55.7 ± 29.4	42.3 ± 30.6 *
During working time, min	515.5 ± 83.9	532.3 ± 75.3
During non-working time, min	326.0 ± 113.0	316.4 ± 97.0
Non-workday, min	679.9 ± 201.8	637.9 ± 178.4
ST during commuting time		
activPAL, min	19.0 ± 17.0	16.4 ± 21.8
OMRON, min	25.4 ± 19.6	18.2 ± 22.7
WLAQ, min	15.4 ± 20.1	9.7 ± 17.2
*p* for difference	<0.01	<0.05
ST during working time		
activPAL, min	408.8 ± 114.9	286.6 ± 160.2 *
OMRON, min	401.7 ± 125.2	280.1 ± 146.3 *
WLAQ, min	456.4 ± 119.5	276.2 ± 175.7 *
*p* for difference	<0.01	0.75
ST during non-working time on workday		
activPAL, min	236.5 ± 79.3	231.8 ± 75.4
OMRON, min	246.0 ± 68.0	247.2 ± 70.6
WLAQ, min	222.2 ± 82.1	225.2 ± 109.7
*p* for difference	<0.01	<0.01
ST on non-workday		
activPAL, min	585.8 ± 133.7	578.8 ± 147.1
OMRON, min	599.1 ± 150.0	595.2 ± 161.4
WLAQ, min	559.7 ± 183.3	485.5 ± 191.7 *
*p* for difference	<0.05	<0.01

Values are presented as mean ± standard deviation. Abbreviation: ST, sitting time; WLAQ, The worker’s Living Activity-time Questionnaire. * Significant differences between sex by Student’s unpaired *t*-tests (*p* < 0.05).

**Table 4 ijerph-18-12774-t004:** Criterion validity of values measured by OMRON and WLAQ compared with acticPAL by the occupation.

	Sedentary Work (*n* = 98)		Activity Work (*n* = 42)	
	ρ	ICC	ρ	ICC
Commuting time				
OMRON	0.74	0.61	0.74	0.83
WLAQ	0.66	0.63	0.67	0.42
Working time				
OMRON	0.84	0.77	0.89	0.87
WLAQ	0.65	0.61	0.79	0.78
Non-working time on workday				
OMRON	0.76	0.71	0.69	0.76
WLAQ	0.45	0.42	0.64	0.59
Non-workday				
OMRON	0.68	0.49	0.66	0.70
WLAQ	0.25	0.24	0.31	0.16

Spearman’s ρ correlations = ρ; Intraclss correlations = ICC. Abbreviation: WLAQ, The worker’s Living Activity-time Questionnaire.

## Data Availability

Raw data were generated at the National Institute of Occupational Safety and Health, Japan (JNIOSH). On reasonable request, derived data supporting the findings of this study are available from the corresponding author, R.S., after approval from JNIOSH and the Research Ethics Committee.

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
