# Peer review of "Validity of Domain-Specific Sedentary Time Using Accelerometer and Questionnaire with activPAL Criterion"

_ijerph, 2021, doi:10.3390/ijerph182312774_

Round 1

Reviewer 1 Report

The paper is well organised, is correctly written, the experimental protocol is well set, the results and discussion are consistent. The conclusion could be improved   The introduction could be improved by adding some more representative references.

Author Response

Dear Reviewer:

The authors thank the reviewer for review of our manuscript and for providing such constructive feedback. The quality of our manuscript has certainly improved as a result of your comments. In response to the editor’s and reviewers’ opinions, we revised our manuscript to show originality and to better define the purpose of our research. Our revisions appear in red in the revised manuscript.

Reviewer 2 Report

The manuscript is written well and in a simple language that makes reading rather easy. 

Objective evaluation of sedentary time is very important in current research in the area of physical inactivity and sedentary behaviour considering the high multitude of data suggesting the clinical implications of sedentary behaviour, but few devices can measure sedentary time accurately. 

I believe, this validity study adds important information to this research area and provides researchers with a good alternative device to be used in working-age populations. 

However, I have some suggestions to improve the manuscript:

  • While I catch the point in giving a background on actigraph in the introduction, I personally wouldn't dedicate too much to the comparison with actigraph in the discussion. The discussion should be used to discuss your results, and omron was not compared to actigraph in this study.
  • page 3, change "recover" to "return"
  • page 3, paragraph 2.3.2: give some more information regarding the  custom-written Excel macro you used
  • page 3, paragraph 2.3.3: change "...time the participants spends..." into "... participants spend"
  • 2.4: "To be eligible, participants needed to wear the accelerometer for at least four workdays and one non-workday, with at least ten 
    hours/day of wear time per day (22)" - please check the following: 1. how is reference 22 appropriate?; 2. leave either hours/day of wear time or hours of wear time per day; 3. in table 3, non-work day wear time appears to range from 4 to 22 hours. Did you include participants wearing accelerometers less than 10 hours/day?
  • Table 3: check the SD of non-workday wearing time. As it is, it appears you included participants who wore accelerometers for 3.5 - 22 hours.
  • page 6: "ME-based", do you mean based on MET-s? If so, it may be more suitable to change to "MET-based.
  • In the discussion, you point out the gender differences in ST, which are very interesting. I would like to see this result written in the "results" section because it passes unobserved in the table.

Author Response

(The authors gave the same response as above.)

Reviewer 3 Report

This is a well-designed, well-written, and valuable study that examines the validity of domain-specific sedentary time measured from accelerometer and questionnaire and using activPAL as a criterion value. I can confirm that the elements necessary for validation study were appropriately included. I would like to extend appreciation to the authors for their contribution to the field of sedentary research. I have added a few minor comments below for the authors to review, however:

(Number of participants)

It may be a misunderstanding on my part, but if 13 shift workers and 67 participants with data missing were excluded from the 332 recruited participants, the total number of participants would appear to be 252.

(Data procedure and statistical analyses, page 4, first paragraph)

It will be easier for readers to understand the selection process if the number of participants excluded from sub-analysis regarding the occupations (and the number of active and sedentary workers) is stated in the text.

(Results, page 4, first paragraph)

Please also describe participants’ average wearing time of activPAL. Did all participants wear it continuously for 24 hours per day, 7 days per week?

(Results, page 4, first paragraph: “The average daily OMRON wearing time of each domain-specific ST period was…” )

Isn't "domain-specific ST period" a mistake of "domain-specific period"?

(Results, Table 4)

Please add the insertion location in Table 4.

Author Response

(The authors gave the same response as above.)
